# α-Phenylalanyl tRNA synthetase competes with Notch signaling through its N-terminal domain

**Manh Tin Ho**, **Jiongming Lu**¤, **Paula Vazquez-Pianzola**, **Beat Suter***

Institute of Cell Biology, University of Bern, Bern, Switzerland

¤ Current address: Max Planck Institute for Biology of Ageing, Köln, Germany
* beat.suter@izb.unibe.ch

## Abstract

The alpha subunit of the cytoplasmic Phenylalanyl tRNA synthetase (α-PheRS, FARSA in humans) displays cell growth and proliferation activities and its elevated levels can induce cell fate changes and tumor-like phenotypes that are neither dependent on the canonical function of charging tRNA$^{Phe}$ with phenylalanine nor on stimulating general translation. In intestinal stem cells of *Drosophila* midguts, α-PheRS levels are naturally slightly elevated and human *FARSA* mRNA levels are elevated in multiple cancers. In the *Drosophila* midgut model, elevated α-PheRS levels caused the accumulation of many additional proliferating cells resembling intestinal stem cells (ISCs) and enteroblasts (EBs). This phenotype partially resembles the tumor-like phenotype described as *Notch* RNAi phenotype for the same cells. Genetic interactions between *α-PheRS* and *Notch* suggest that their activities neutralize each other and that elevated α-PheRS levels attenuate Notch signaling when Notch induces differentiation into enterocytes, type II neuroblast stem cell proliferation, or transcription of a Notch reporter. These non-canonical functions all map to the N-terminal part of α-PheRS which accumulates naturally in the intestine. This truncated version of α-PheRS (α-S) also localizes to nuclei and displays weak sequence similarity to the Notch intracellular domain (NICD), suggesting that α-S might compete with the NICD for binding to a common target. Supporting this hypothesis, the tryptophan (W) residue reported to be key for the interaction between the NICD and the Su(H) BTD domain is not only conserved in α-PheRS and α-S, but also essential for attenuating Notch signaling.

## Author summary

Aminoacyl tRNA synthetases charge tRNAs with their cognate amino acid to ensure proper decoding of the genetic code during translation. Independent of its aminoacylation function, the alpha subunit of *Drosophila* cytoplasmic Phenylalanyl tRNA synthetase (α-PheRS, FARSA in humans) has an additional activity that promotes growth and proliferation. Here we describe that elevated α-PheRS levels also induce cell fate changes and tumorous phenotypes in *Drosophila* midguts. Excessive proliferating cells with stem and progenitor cell characteristics accumulate and the composition of the terminally

**Data Availability Statement:** All relevant data are within the manuscript and its Supporting Information files.

**Funding:** This work was financially supported by the University of Bern (https://www.unibe.ch) to B.

S., by the project grants 31003A_173188 and 310030_205075 from the Swiss National Science Foundation (SNF; https://www.snf.ch) to B.S., and by a grant from the Novartis Foundation for Medical-Biological Research (#18A050) to B.S. Equipment support was by an equipment grant from SNF (316030_150824) to B.S. and by the University of Bern (https://www.unibe.ch). The funders had no role in study design, data collection and analysis, decision to publish, or preparation of the manuscript.

**Competing interests:** The authors have declared that no competing interests exist.

differentiated cells changes, too. This phenotype together with observed genetic interactions between α-PheRS and Notch levels show that α-PheRS counteracts Notch signaling in many different tissues and developmental stages. This novel activity of α-PheRS maps to its N-terminal part, which is naturally produced. The fragment contains a DNA binding domain, translocates into nuclei, and displays essential similarities to a Notch domain that binds to the downstream transcription factor. This suggests that it might be competing with Notch for binding to a common target. Not only because Notch plays important roles in many tumors, but also because *FARSA* mRNA levels are considerably upregulated in many tumors, this novel activity deserves more attention for cancer research.

## Introduction

Aminoacyl-tRNA synthetases (aaRSs) are essential enzymes that act by charging transfer RNAs (tRNAs) with their cognate amino acid, a key process for protein synthesis [1]. Besides their well-known role in translation, aaRSs perform additional functions in the cytoplasm, the nucleus, and even outside of the cell [2–10]. Phenylalanyl-tRNA synthetase (PheRS, FARS, or FRS) displays elevated expression levels in many cancers compared to their healthy counterparts according to the database "Gene Expression across Normal and Tumor tissue" (GENT2; [11]). Such a correlation had been reported already two decades earlier and it was suggested that an alternative activity of PheRS might contribute to the tumorigenic events [12–13]. A moonlighting function of the α subunit of *Drosophila* PheRS has recently been found to promote growth and proliferation in different tissues [14]. Despite this, a possible mechanism explaining the connection between elevated PheRS levels and tumor formation had so far not been reported and, to our knowledge, also not been studied.

Cytoplasmic PheRS is the most complex member of the aaRSs family, a heterotetrameric protein consisting of 2 alpha- (α) and 2 beta- (β) subunits responsible for charging tRNA$^{Phe}$ for translation [15]. The α subunit includes the catalytic core of the tRNA synthetase and the β subunit has structural modules with a wide range of functions, including tRNA anticodon binding, hydrolyzing misactivated amino acids, and editing misaminoacylated tRNA$^{Phe}$ species [15–17]. Importantly, both subunits are needed for aminoacylation of tRNA$^{Phe}$.

We set out to address the question of whether and how the elevated levels of α-PheRS that induce growth and proliferation [14] might contribute to tumor formation. To test for this activity, we studied the role of α-PheRS levels in the *Drosophila* model system, in which dissecting the molecular mechanism of such a moonlighting role of *α-PheRS* seemed feasible. Tissue growth and homeostasis play important roles in developing and outgrown animals, and they require tight control of stem cell self-renewal and differentiation into daughter cells. The *Drosophila* midgut is a powerful model to analyze these mechanisms and their interplay. Adult intestinal stem cells (ISCs) and adult midgut progenitors (AMPs) in the larval gut can either divide asymmetrically or symmetrically [18,19]. Asymmetric divisions of ISCs give rise to a new ISC and a Su(H)GBE$^+$ enteroblast (EB) for the enterocyte (EC) lineage differentiation, or a pre-enteroendocrine cell (pre-EE) for the enteroendocrine (EE) cell lineage differentiation [20,21]. Differential Delta/Notch signaling between the ISC and its daughter causes the latter to either differentiate into an absorptive EC lineage or a secretory EE lineage by distinct mechanisms [22–25]. EBs are intermediate differentiating cells that differentiate into ECs in a Notch-dependent manner [26], while the production of EEs has not been molecularly characterized. However, the pre-EEs require only low levels of Notch signaling to differentiate into EEs [21]. Intestinal homeostasis is an interesting system to elucidate the control of continuous replenishment of lost cells and the maintenance of stem cells. Additionally, it turned out to be

also a very dynamic process that can react to the loss of large numbers of differentiated cells in response to injuries and bacterial infections.

We found that in different tissues and cell types, α-PheRS levels in stem and progenitor cells regulate cell proliferation, cell differentiation, or both. Interestingly, a proteolytic α-PheRS fragment that can neither interact with β-PheRS nor perform aminoacylation, promotes these growth, proliferation, and differentiation functions. Although the consequences of altered levels vary to some degree between tissues, we found that even in tissues with the most divergent consequences, α-PheRS levels act by attenuating Notch signaling. Sequence comparisons combined with genetic and biochemical tests, as well as the cellular localization of this truncated α-PheRS (α-S) point to a mechanism whereby the N-terminal "DNA binding domain" of α-PheRS might compete with the NICD for the binding to a common target.

## Results

### PheRS is enriched in *Drosophila* gut stem cells

PheRS promotes growth and proliferation in different tissues [14]. In the *Drosophila* midgut, intestinal stem cells (ISCs) display very high proliferation rates [27] due to a high turnover of the cells with a nutrient absorption mission. In addition, the GENT2 database also reports highly elevated PheRS levels in many malignant cells compared to normal tissues, and among these are also intestinal cancers [11]. To investigate the role of PheRS in intestinal cells, we examined the PheRS expression levels in wild-type fly guts. ISCs and AMPs are ideal to study the novel PheRS activity because they can both undergo either symmetric or asymmetric divisions and differentiate to maintain the cell population in the fly gut. We observed that adult midguts express naturally elevated levels of α-PheRS in the diploid wild-type cells compared to the enterocytes in the same tissue (Fig 1A–1A", and 1B). Similarly, a Myc-tagged α-PheRS under the normal genomic *α-PheRS* promoter is expressed at higher levels in AMPs than in enterocytes of larval guts (Fig 1C–1C" and 1D). To compare the natural levels of α-PheRS in different cell types of the adult midgut, we used the *esg-Gal4*, *tub-Gal80*$^{ts}$, *UAS-2XeYFP* system (hereby we refer as esg$^{ts}$) to label ISCs and EBs [22] and Prospero staining to label enteroendocrine cells (EEs). This setup revealed stronger staining for endogenous α-PheRS in EEs (Pros$^+$) than in enterocytes (ECs; Fig 1E–1F""). In the same assay, the narrow cytoplasm of the YFP$^+$ progenitor cells shows only a slightly higher pixel intensity (white arrows).

### Elevated α-PheRS levels promote ISC and EE accumulation and altered EB morphology

We previously constructed a mutant *α-PheRS* gene, that encodes an aminoacylation-dead subunit (*α-PheRS*$^{Cys}$) that still displays the non-canonical activity of promoting growth and proliferation [14]. This mutant allows us to specifically study the activities of *α-PheRS* that are not dependent on the aminoacyl tRNA synthetase activity of PheRS. To investigate the effect of elevated levels of α-PheRS and α-PheRS$^{Cys}$ (α-PheRS$^{(Cys)}$) on adult midgut ISCs and EBs, we used *α-PheRS*$^{(Cys)}$ under UAS control together with the esg$^{ts}$ driver system [22]. Co-expression of *UAS-2XEYFP* allowed us to monitor the activity of the *esg-Gal4* driver and to label ISCs and EBs. Upon controlled upregulation of α-PheRS$^{(Cys)}$ levels, we observed two phenotypes in the intestines as early as 2 days after induction and the phenotypes became more pronounced until day 5 (S1 Fig). The first phenotype resembled partially a *Notch* (*N*) knockdown-like phenotype with a 4.3 and 4.7-fold, respectively, increase in the total numbers of YFP$^+$ cells upon additional expression of α-PheRS and α-PheRS$^{Cys}$, respectively (Fig 2A–2C"' and 2G). The increase of these midgut progenitor cells (ISCs and EBs) could be a consequence of increased

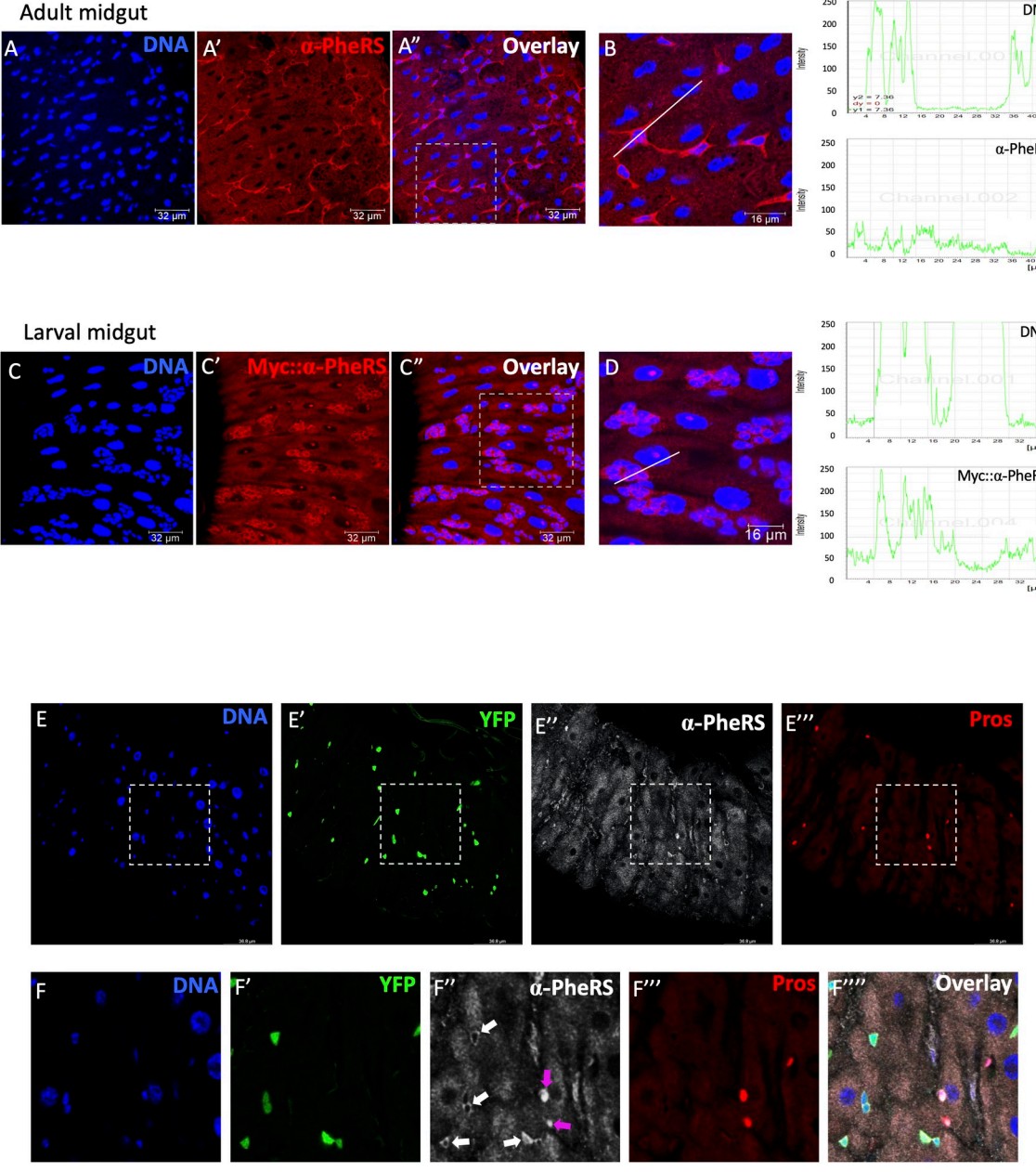

**Fig 1. α-PheRS is enriched in *Drosophila* progenitor and enteroendocrine cells.** (A-A", B, C-C", D) Endogenous α-PheRS accumulation is higher in small, diploid progenitor cells in wild-type adult and larval midguts. Signal intensity values in the graphs (right column) were measured from left to right along the lines in B and D. (E-F"") The *esg-Gal4,UAS-2XEYFP;tub-Gal80ᵗˢ* (= esgᵗˢ) system and Prospero staining, respectively, were used to label progenitor and enteroendocrine cells, respectively. Animals were mated at 18°C and adults of the required genotypes were collected and shifted to 29°C to inactivate Gal80ᵗˢ. Adult midguts were dissected from female flies after 5 days of induction.

proliferation and/or a differentiation problem of ISCs [28,29]. Indeed, we also observed an increase in the proportion of EEs from 8% in the control to 12% in the posterior midguts with elevated *α-PheRS* expression (Fig 2H), indicating that elevated expression of *α-PheRS* promotes ISC proliferation and EE accumulation.

The controlled upregulation of *α-PheRS^(Cys)* levels also led to a second phenotype, an altered morphology of EBs. We observed a hyperaccumulation of YFP⁺ polyploid cells that resembled

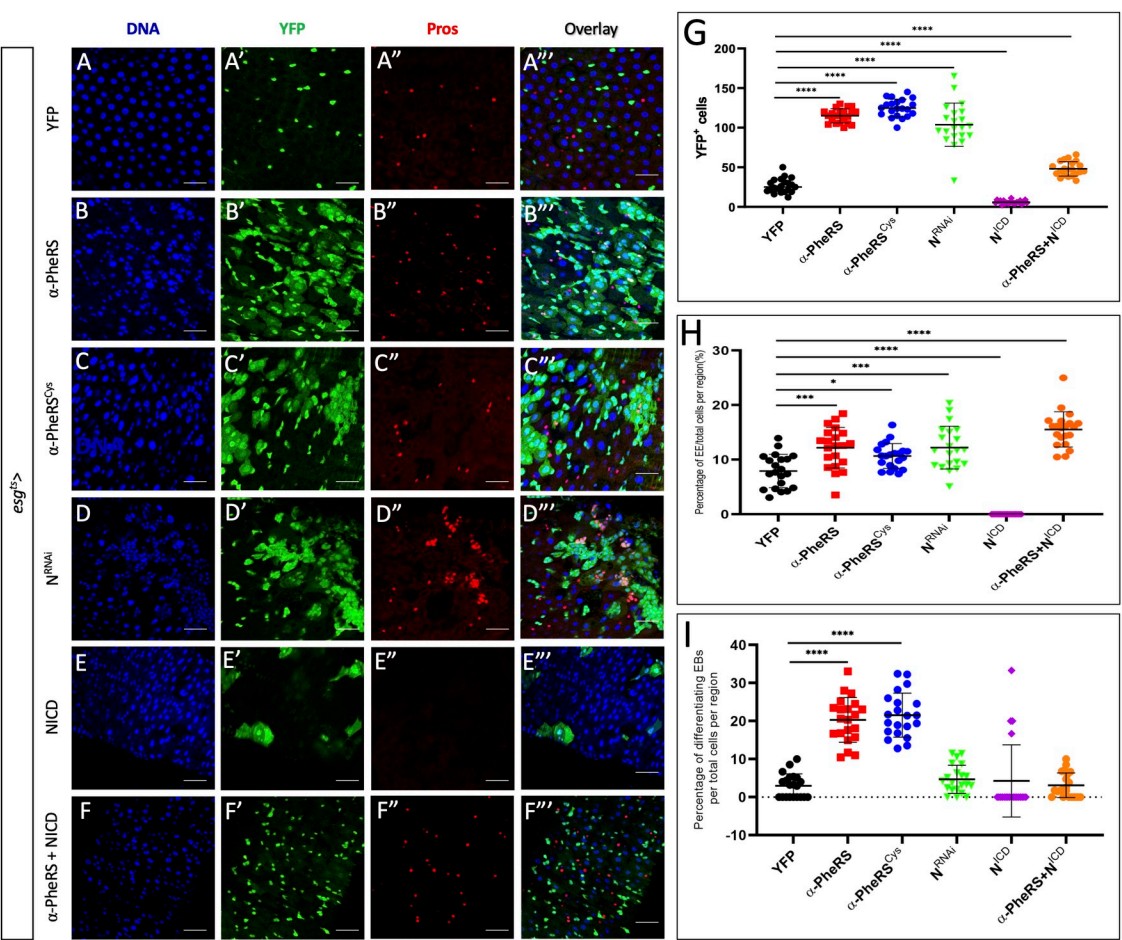

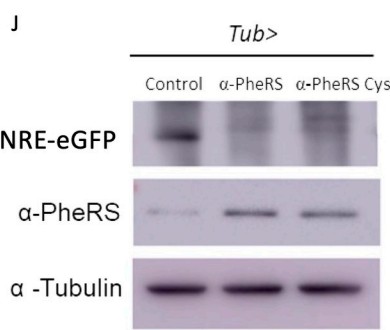

**Fig 2. Elevated α-PheRS levels affect gut homeostasis, leading to additional progenitor cells, EEs, and differentiating EBs.**
(A-C''') Elevated α-PheRS or α-PheRS$^{Cys}$ levels induced by the esg$^{ts}$ system (see Fig 1E for details) led to additional YFP-positive cells and also to the accumulation of EEs *(esg$^{ts}$/+;UAS-α-PheRS$^{(Cys)}$/+)*. (D-D''') RNAi knockdown of *Notch (N)* with the same esg$^{ts}$ system led to a similar phenotype as elevation of α-PheRS levels *(UAS-N$^{RNAi}$/+; esg$^{ts}$/+)*. (E-F''') Elevation of α-PheRS levels rescued the depleted ISC pool caused by Notch over-activation (induced by expressing the NICD under esg$^{ts}$ control). (G) Quantification of YFP$^+$ cell numbers. (H) Proportional contribution of EEs to the total number of gut cells in the intestinal region analyzed. (I) Percentage of differentiating EBs among YFP$^+$ cells. In all experiments, EEs were visualized with the anti-Prospero antibody, YFP$^-$/Hoechst$^+$ polyploid cells were counted as ECs. Animals were mated at 18˚C, and adults of the required genotypes were collected and shifted to 29˚C to inactivate Gal80$^{ts}$. Adult midguts were dissected from female flies after 5 days of induction at 29˚C. At least 10 guts were analyzed for each genotype. n = 10, $^{**}p<0.01$, $^{***}p<0.001$, $^{****}p<0.0001$ in ANOVA tests. (J) The Notch activity reporter (NRE-eGFP) expresses GFP under the control of a Notch response element. Its levels drastically decreased upon elevated expression of *α-PheRS* or *α-PheRS$^{Cys}$* in adult midgut ISCs *(esg-Gal4,UAS-2XEYFP/+;tub-Gal80$^{ts}$/UAS-α-PheRS$^{(Cys)}$,NRE-eGFP)*. Female flies were collected 3 days after eclosure and cultured at 29˚C for 5 days before dissecting and harvesting their midguts.

the ones that have been referred to in other studies as "differentiating" EBs [30,31]. Because these cells display ISC/EB characteristics (YFP+) and EC characteristics (large polyploid nuclei) at the same time, it appears that they arrested development between these two fates. Differentiating EBs were rarely observed in wild-type midguts (3% of progenitor cells; Fig 2A–2A''' and 2I) and if they were present, they retained only weak YFP signals. In midguts expressing elevated levels of α-PheRS$^{(Cys)}$, we observed much higher levels of "differentiating" EBs (20% and 22%, respectively, of the YFP+ progenitor cells), and these cells also displayed a strong YFP signal (Fig 2B–2C''' and 2I). We concluded that the elevated levels of α-PheRS in the intestine induce hyperaccumulation of ISCs and EEs and alterations in EB morphology.

## The phenotype of elevated α-PheRS levels partially resembles the low Notch phenotype

Notch signaling is an important signaling pathway that coordinates ISC proliferation and differentiation [26]. We noted that the above-described changes regarding ISCs and EEs partially overlapped with the low $N$ activity phenotype that had been reported in several studies [22,23,32–34]. To confirm this similarity, we knocked down $N$ with RNAi treatment with the same esg$^{ts}$ system and this, indeed, led to a typical phenotype in ISC proliferation and EE accumulation as reported in the studies mentioned above (Fig 2D–2D'''). This allowed us to compare the $N^{RNAi}$ phenotype with the phenotype of elevated $\alpha$-PheRS$^{(Cys)}$ levels (Fig 2B–2C'''). The increase in YFP+ cells (290%; from 27 to 104 cells) of $N^{RNAi}$ treatment was in the same range as the one observed under elevated $\alpha$-PheRS expression (332%; from 27 to 115 cells), and with 12%, the proportion of EEs was the same under the two conditions (Fig 2G and 2H). We also used the esg$^{ts}$ driver to upregulate Notch signaling by expressing the continuously active form of Notch, the Notch intracellular domain (NICD). This caused ISCs to differentiate into ECs, thereby depleting 80% of the ISC pool (Fig 2E–2E''' and 2G) and inhibiting EE differentiation (Fig 2E–2E''' and 2H) as it has been reported in previous studies [22,31].

To test whether α-PheRS$^{(Cys)}$ interferes with Notch signaling, we simultaneously elevated levels of $\alpha$-PheRS$^{(Cys)}$ and NICD. This schema had a rescuing effect on both phenotypes. On the one hand, it rescued the ISC pool and the proportion of EEs depleted by NICD expression even beyond the wild-type levels (Fig 2F–2F''' and 2G). On the other hand, the phenotype caused by the additional expression of the NICD can also be interpreted as a partial rescue of the phenotype caused by elevated α-PheRS$^{(Cys)}$ levels. The numbers of YFP+ cells and differentiating EBs dropped considerably, even though they remained still higher than in the wild type (Fig 2D–2F''' 2G and 2I). These results suggest that the two phenotypes observed when $\alpha$-PheRS$^{(Cys)}$ expression was elevated in ISCs and EBs arise because $\alpha$-PheRS$^{(Cys)}$ and Notch signaling counteract each other for the control of proliferation and differentiation.

We next tested whether α-PheRS$^{(Cys)}$ can downregulate $Notch$ signaling in adult midguts using the $Notch$ reporter NRE-eGFP (Notch response element promoter driving the expression of eGFP) [35]. As seen in Fig 2J, the expression of α-PheRS and α-PheRS$^{Cys}$ clearly reduced the $N$-driven expression of the N-signaling reporter. We conclude that elevated α-PheRS levels in midgut stem cells cause misregulation of gut homeostasis and that this effect is not only the consequence of the proliferative role of α-PheRS but also caused by downregulating Notch signaling and inhibiting the terminal differentiation of the progenitor cells (EBs).

## Elevated α-PheRS levels induce AMP proliferation and interfere with gut homeostasis

Larval *Drosophila* midguts possess AMPs as the temporary reserve for adult ISCs. AMPs go through several divisions to form imaginal midgut islets that consist of AMPs enclosed by

peripheral cells (PC) [36]. Due to the transient period of the larval stage, larval midguts do not require EC renewal under normal conditions [37]. However, larval guts can respond to certain damages and injuries by triggering a regenerative process that compensates for depleted ECs in the epithelium and also allows reconstituting the AMP pool [38]. We also investigated the effect of elevated *α-PheRS* levels on larval AMPs by using the *esg-Gal4*, *UAS-2XEYFP* driver, which is specifically expressed in AMPs. Co-expression of *UAS-2XEYFP* allowed us to monitor the activity of the *esg-Gal4* driver and to label AMPs. The results demonstrated that elevated Myc::α-PheRS in the larval midgut significantly increased the proportion of EE and PH3 positive cells (mitotic cells) in the posterior midgut (Fig 3A–3D and 3I–3M). This increased EE proportion again pointed out the similarity with the Notch knockdown-like phenotype in the adult gut. Driving the expression of α-PheRS$^{Cys}$ in AMPs also produced a higher mitotic index (Fig 3E–3F and 3L), indicating that the aminoacylation function is not required for this activity. Due to the lack of epithelial renewal in normal larval guts, the cell density of larval midguts is usually stable [36,37]. It is therefore surprising that elevated α-PheRS$^{(Cys)}$ levels increase the cell density in larval midguts (Fig 3N). Altogether, these results show that elevated α-PheRS$^{(Cys)}$ levels cause AMPs to proliferate and to differentiate, producing additional larval midgut cells.

Strikingly, elevated α-PheRS$^{Cys}$ levels induced tumor-like phenotypes in both anterior and posterior areas of the larval midgut (outlined with white dashed lines, Fig 3E and 3F) while the elevation of wild-type Myc::α-PheRS only gave rise to high numbers of AMPs in the posterior larval midgut (Fig 3C and 3D). Elevated α-PheRS$^{Cys}$ levels also caused the appearance of a more severe tumor-like phenotype, where individual AMP islets could not be discerned anymore (Fig 3E and 3F). Wild-type larval guts contain ECs with large nuclei and interspersed occasional AMP islets with smaller nuclei (as seen in the YFP overexpression control, Fig 3A and 3B). In contrast, we observed a phenotype where ECs and AMPs could not be distinguished based on the size of their nuclei but emerged as a larger cell population with intermediate size nuclei (Fig 3D). Many of these cells expressed the *esg*>YFP stem cell marker at high levels, but others displayed only a very weak YFP signal. This result is comparable to the EB phenotype in adult midguts where the progenitor cells remain in an intermediate differentiation state (Fig 2B''' and 2C'''). Notably, this phenotype was also observed when Notch was downregulated by RNAi using the same induction system (Fig 3G and 3H). This points again to the possibility that downregulation of Notch signaling is involved in producing this phenotype.

## *α-PheRS* is a novel general repressor of Notch signaling

Developing larval brains contain neuroblasts (NBs), neuronal stem cells that divide asymmetrically with one daughter keeping the stemness and the other differentiating into a neuronal cell. Notch signaling plays a crucial role in renewing type II NBs [39,40], but in contrast to the situation in the gut, loss of *Notch* prevents type II NB self-renewal, whereas ectopic expression of activated Notch leads to tumor formation [41–43]. This opposite role of *Notch* makes the type II NB lineage an ideal complementary system to test whether α-PheRS is a general component of the *Notch* pathway. Driving α-PheRS or the α-PheRS$^{Cys}$ expression in NBs with the *inscutable-Gal4 (incs-Gal4)* driver resulted in significantly smaller central brains (CB), the region where the type I and II NBs are located (Fig 4A–4C and 4G). In contrast, this treatment had little or no effect on the size of the optic lobes (OL). Importantly, the phenotype was virtually indistinguishable from the phenotype caused by *N* knock-down (Fig 4E), although the phenotype of the latter might show slightly higher expressivity. Analogous to the situation in the gut, co-expression of NICD together with *α-PheRS$^{Cys}$* in neuroblasts rescued the brains to wild-type size (Fig 4D and 4G).

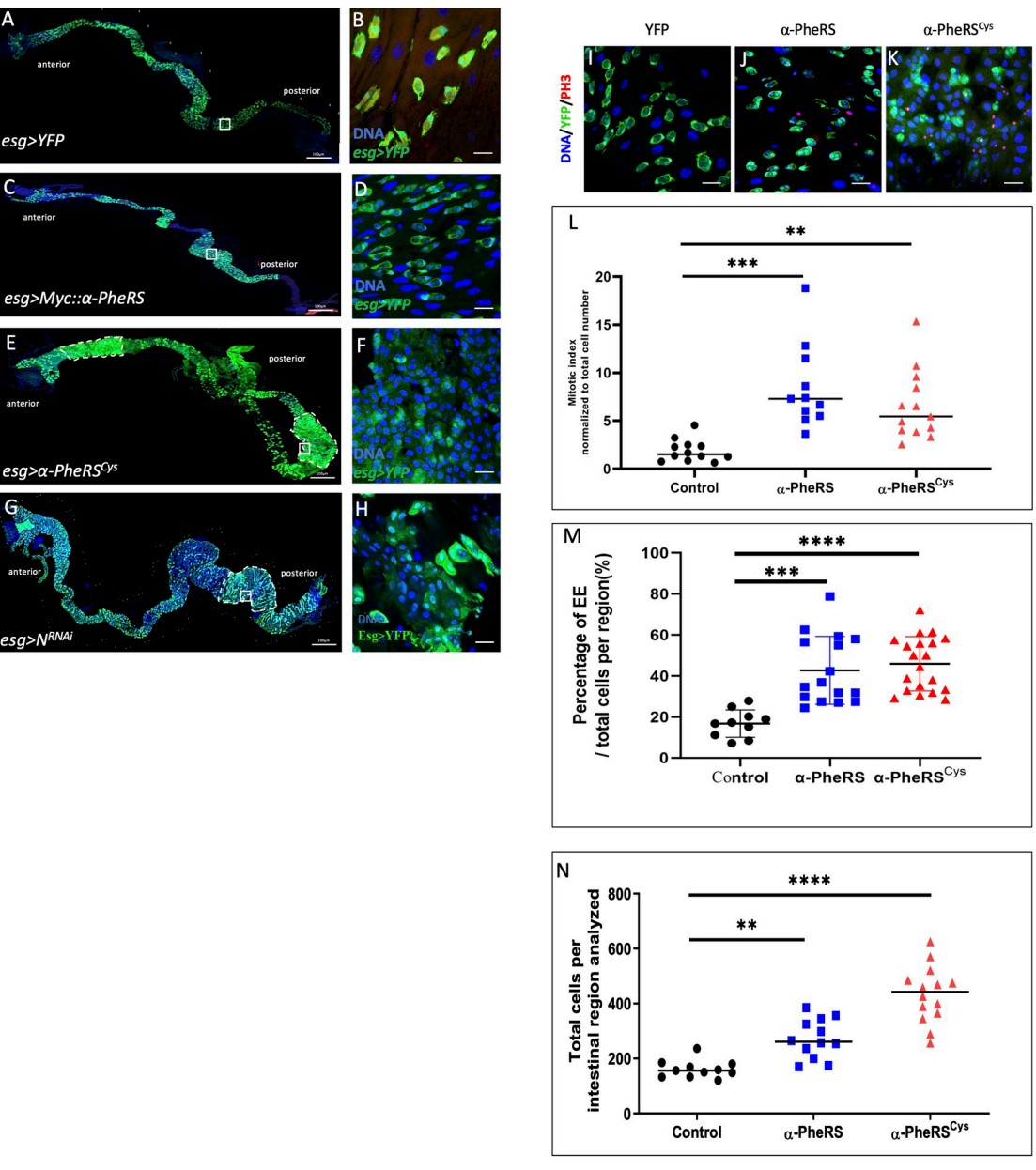

**Fig 3. The phenotype of elevated α-PheRS levels resembles partially a *Notch*-like phenotype in larval guts.** (A-H) Whole gut images and high-power micrographs of the posterior midgut area marked with a white square in the overview picture. The expression of the different genes indicated was driven with the *esg-Gal4,UAS-2XEYFP* system. The YFP signal displayed in green marks AMPs and EBs, Hoechst (blue) marks the nuclear DNA. (E, F) Elevated *α-PheRS^{Cys}* expression gave rise to tumor-like areas in both the anterior and the posterior midgut (outlined with dashed lines) (*esg-Gal4,UAS-2XEYFP/+;UAS-α-PheRS^{(Cys)}/+*). (I–M) Mitotic cells were identified with anti-phospho-Histone H3 (PH3) antibodies (red channel) and the PH3+ cells are counted and normalized to the total cell number per intestinal region. n = 10, **p<0.01, ***p<0.001 in t-tests. EEs were identified based on their morphology and their relative abundance is shown in (M). (N) Total cells per intestinal region (total cells per frame of 63x objective of confocal microscope with 1024x1024 pixels) were measured by counting Hoechst 33258 labeled cells manually. n = 10, **p<0.01,****p<0.0001.

Type II neuroblasts are particularly suited to analyze effects on neuroblast differentiation. Targeting specifically the eight type II neuroblasts in central brain lobes with ectopic *α-PheRS* or *α-PheRS^{Cys}* expression, resulted in a strong reduction of the number of type II neuroblasts (Fig 4H–4J and 4N). Knocking down *Notch* in type II NBs with RNAi and the same driver

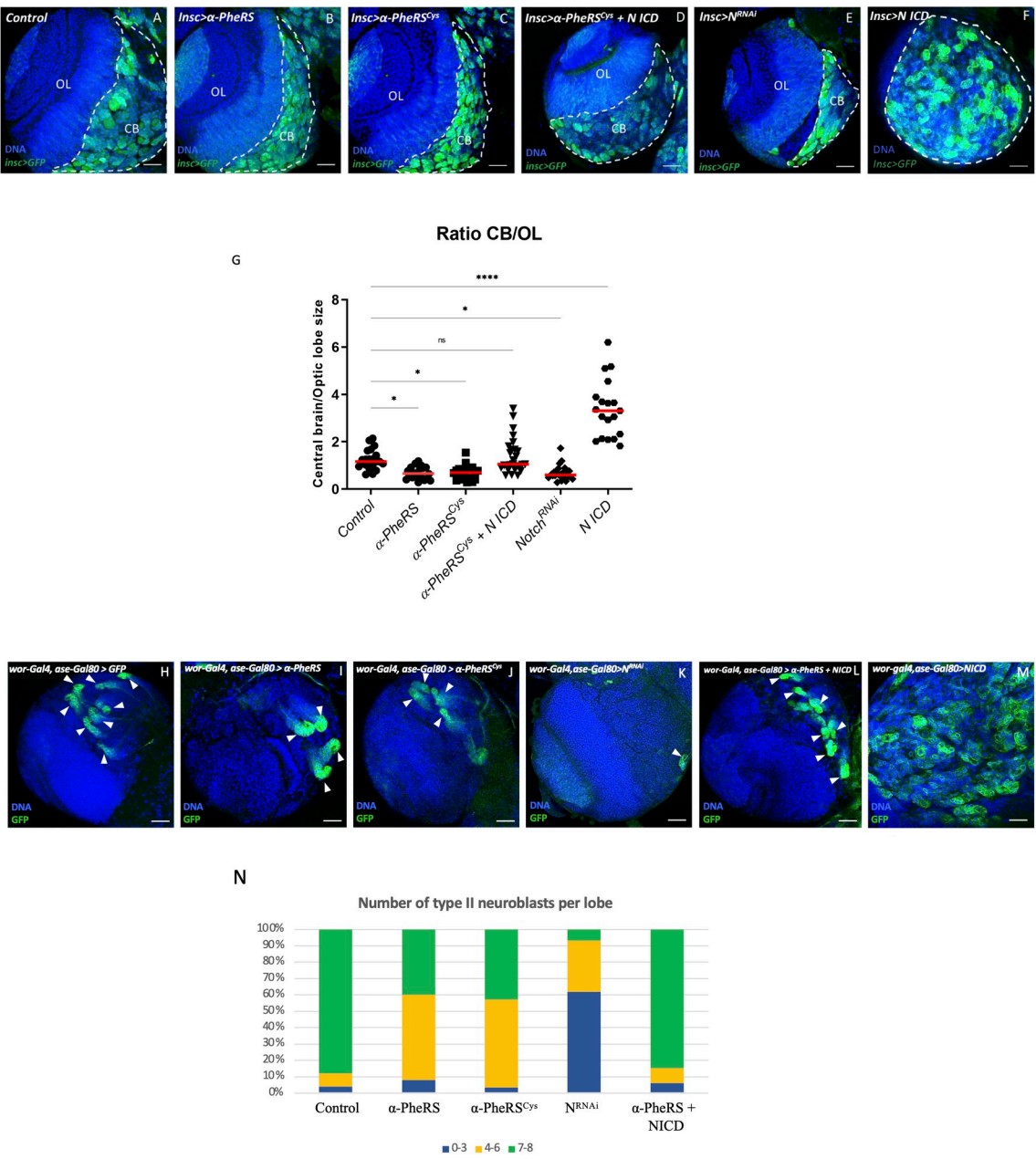

**Fig 4. $\alpha$-PheRS counteracts Notch activity in neuroblast proliferation.** (A-F) Additional expression of $\alpha$-PheRS or $\alpha$-PheRS$^{Cys}$ in neuroblasts reduced the ratio of central brain (CB) size to optic lobe (OL) size. The *insc-Gal4,UAS-GFP* system was used to drive the expression of the test genes in neuroblasts (NBs) (*w*$^*$, *Insc-Gal4/UAS-GFP;UAS-$\alpha$-PheRS$^{(Cys)}$/+*). NBs were labeled by GFP expression. CBs are shown on the right side (area with GFP$^+$ NBs and outlined by white dashed lines) and the OL area on the left. N knock-down (E) and NICD overexpression, alone (F) and in combination with $\alpha$-PheRS (D), are also shown. (G) Effect of increased expression of $\alpha$-PheRS$^{(Cys)}$ and *NICD*, and of *N* knockdown on the CB/OL size ratio. n = 20, $^{***}$p<0.001, *ns*: not significant. (H-M, N) Elevated $\alpha$-PheRS or $\alpha$-PheRS$^{Cys}$ levels reduced the number of Type II NBs per brain lobe, and less than half of the lobes contained the normal 7–8 NBs. *Notch* RNAi caused the same phenotype with higher expressivity. Additional expression of $\alpha$-PheRS$^{(Cys)}$ rescued the tumor phenotype caused by ectopic Notch activation in NB to normal wild-type levels. The larval brains were dissected from third instar larvae, and at least 20 brains were analyzed for each genotype (*w*$^*$,*UAS-Dicer2/+;wor-Gal4,ase-Gal80/+;UAS-mCD8::GFP/UAS-$\alpha$-PheRS$^{(Cys)}$*). Each brain lobe was classified according to the number of Type II NBs per brain lobe. n = 25.

combination resulted in a similar phenotype, but with a higher expressivity (Fig 4K, 4L, 4N). On the other hand, overexpression of α-PheRS in type II neuroblasts where Notch signaling was also over-activated by the co-expression of NICD (Fig 4L) partially rescued the tumor phenotype of type II neuroblasts to wild-type numbers and it restored the normal size of the brain (Fig 4L). Because the genetic interaction with *Notch* shows that the two activities counteract one another in guts and brains, two tissues where *Notch* has opposing functions on cellular differentiation, it appears that the interaction between *α-PheRS* and *Notch* is not mediated indirectly through a primary effect on division or differentiation but is likely to be a more direct one. In any case, this interaction appears to modulate tissue homeostasis in different organs.

To further address the role of *α-PheRS* on Notch signaling, we also studied its effect on the larval wing disc and the adult wing. Using the *en-Gal4, tub-Gal80$^{ts}$* (referred to as en$^{ts}$ system) to elevate *α-PheRS* expression allowed us to specifically and conditionally activate *α-PheRS$^{(Cys)}$* expression in the posterior compartment of the wing disc by temperature-shift control. Reduced Notch signaling in the posterior wing disc can cause the notched wing phenotype at the distal end of the wing and the emergence of an ectopic partial vein in the center of the posterior compartment. The additional expression of *α-PheRS$^{(Cys)}$* in the posterior compartment did not produce the distal phenotype and we will discuss possible reasons for this in the Discussion. However, elevated *α-PheRS* or *α-PheRS$^{Cys}$* levels induced the central ectopic vein phenotype also observed under low Notch signaling conditions (S4A–S4D Fig) [44]. The new vein branches off from the connecting vein between the L4 and L5 vein (arrowhead, S4B and S4C Fig) and is indistinguishable from the one obtained by downregulating *Notch* by RNAi treatment using the same expression system (S4D Fig). Again, these findings suggest that *α-PheRS* might be a more general, novel regulator of Notch signaling.

To test directly for changes in *Notch* activity in response to altered levels of *α-PheRS$^{(Cys)}$* in imaginal wing discs, we monitored Notch activity with the NRE-eGFP reporter and used the en$^{ts}$ system together with *UAS-mRFP* to conditionally express *α-PheRS$^{(Cys)}$* in the posterior compartment. In this system, eGFP expression marks the cells with active Notch signaling, which normally localize along the dorsal-ventral (D/V) boundary of the larval wing disc (Fig 5A"), and it marks the posterior, en-Gal4$^{+}$, compartment of the wing disc with mRFP, allowing us to assess *N* activity in this compartment (black arrowhead, Fig 5A"') while using the anterior compartment as the internal control. In wild-type wing discs, the NRE-eGFP signal is as strong in the posterior compartment as in the anterior one (Fig 5B–5B"). Knockdown of *Notch* in the posterior compartment resulted in the exclusion of the NRE-eGFP signal from the mRFP-marked posterior compartment (white arrow, Fig 5E–5E"'). Importantly, elevated *α-PheRS$^{(Cys)}$* levels also caused a loss of the eGFP signal in the mRFP-marked posterior compartments (Fig 5C–5C"' and 5D–5D"'). These observations demonstrate that elevated *α-PheRS* levels downregulate *Notch* activity in larval wing discs and that this inhibition acts on Notch signaling at or before the transcription of the reporter. Taken together, the results from midguts, brains, and wing discs demonstrate that *α-PheRS* is a novel general repressor of *Notch* signaling.

## The N-terminal 200 amino acids of α-PheRS (α-S) are sufficient to induce the proliferative phenotype and to repress Notch activity

Having shown that the growth and proliferation function and the counteracting of the Notch signaling are independent of the aminoacylation activity of α-PheRS, we wanted to find out whether this activity can be separated from the catalytic domain. As shown in Fig 6A and 6B, the C-terminal catalytic domain is separated by a linker region from the N-terminal part. We, therefore, cloned the first 200 codons of the α-PheRS ORF under UAS control and added a Myc tag at the beginning of the open reading frame, giving rise to the *UAS-Myc::α-S (α-S)*

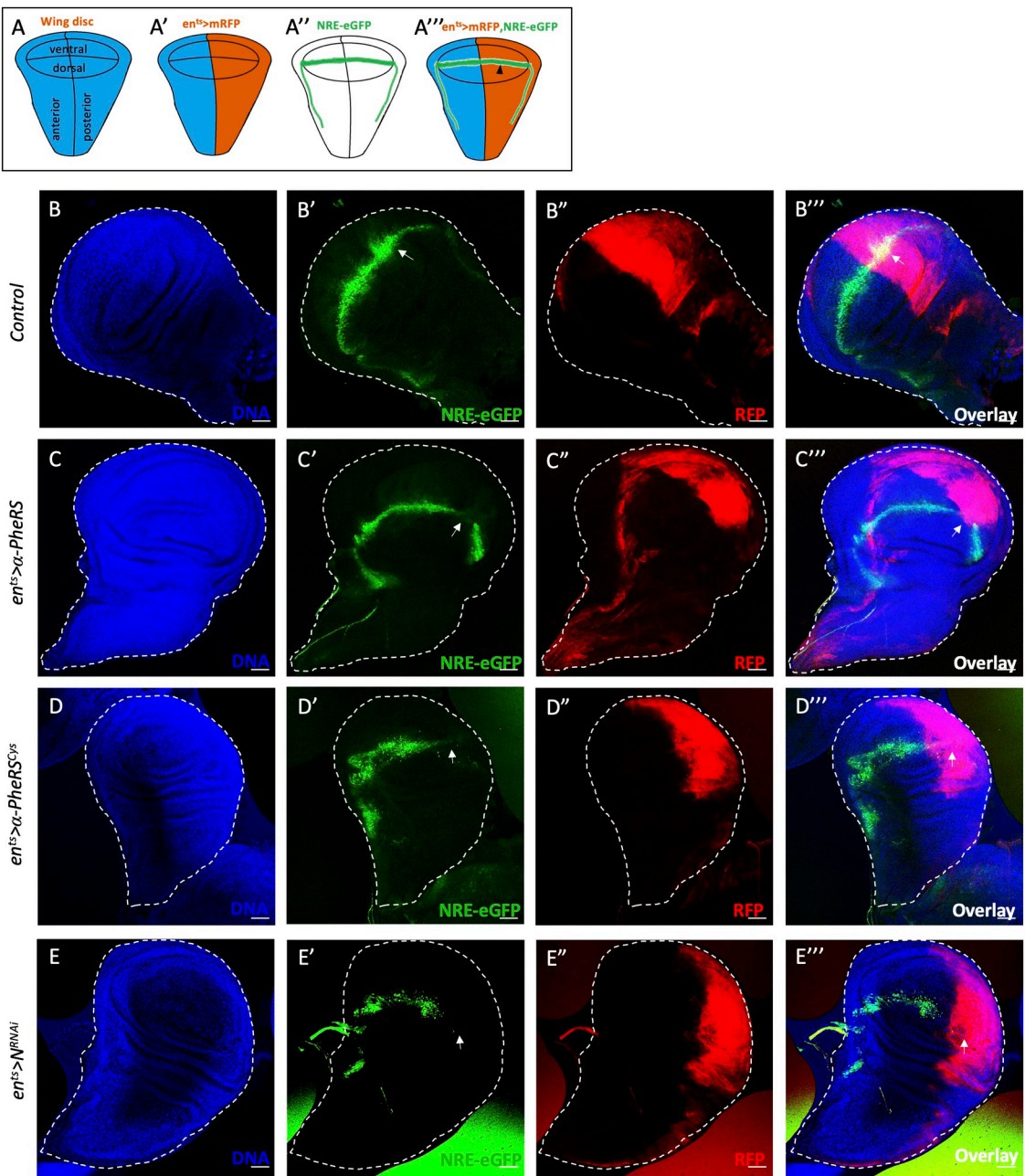

**Fig 5. Elevated α-PheRS levels repress Notch signaling in wing discs.** Schematic illustration of wild-type wing imaginal discs showing the position of the different compartments (A), the expression domain of en^ts>mRFP (A'), and the expression domain of *NRE-eGFP* (A") along the dorsal/ventral (D/V) boundary. (A''') *en-Gal4,NRE-eGFP,UAS-mRFP;tub-Gal80^ts* was used to drive transgene expression in the posterior compartment of developing wing discs and to evaluate the Notch activity with the reporter *NRE-eGFP* (black arrowhead) *(w,UAS-Dcr2/+;en-Gal4,UAS-mRFP,NRE-eGFP/+;tub-Gal80^ts/UAS-α-PheRS^(Cys))*. (B-E"') Elevated α-PheRS or α-PheRS^Cys led to the loss of NRE-eGFP expression in the posterior compartment, similar to the phenotype of *Notch* knockdown *(w,UAS-Dcr2/UAS-N^RNAi;en-Gal4,UAS-mRFP,NRE-eGFP/+;tub-Gal80^ts/+)*. After egg-laying and incubation at 18°C for 6 days, the animals were shifted to 29°C for 1 day to inactivate Gal80^ts and to enable expression of α-PheRS, α-PheRS^Cys, or N^RNAi.

gene (Fig 6B). Expressing *α-S* with the en-Gal4 driver and staining wing discs for Myc, revealed that this truncated version of α-PheRS was stably expressed in the posterior wing disc compartment (S2 Fig). To test whether this C-terminally truncated isoform was able to repress

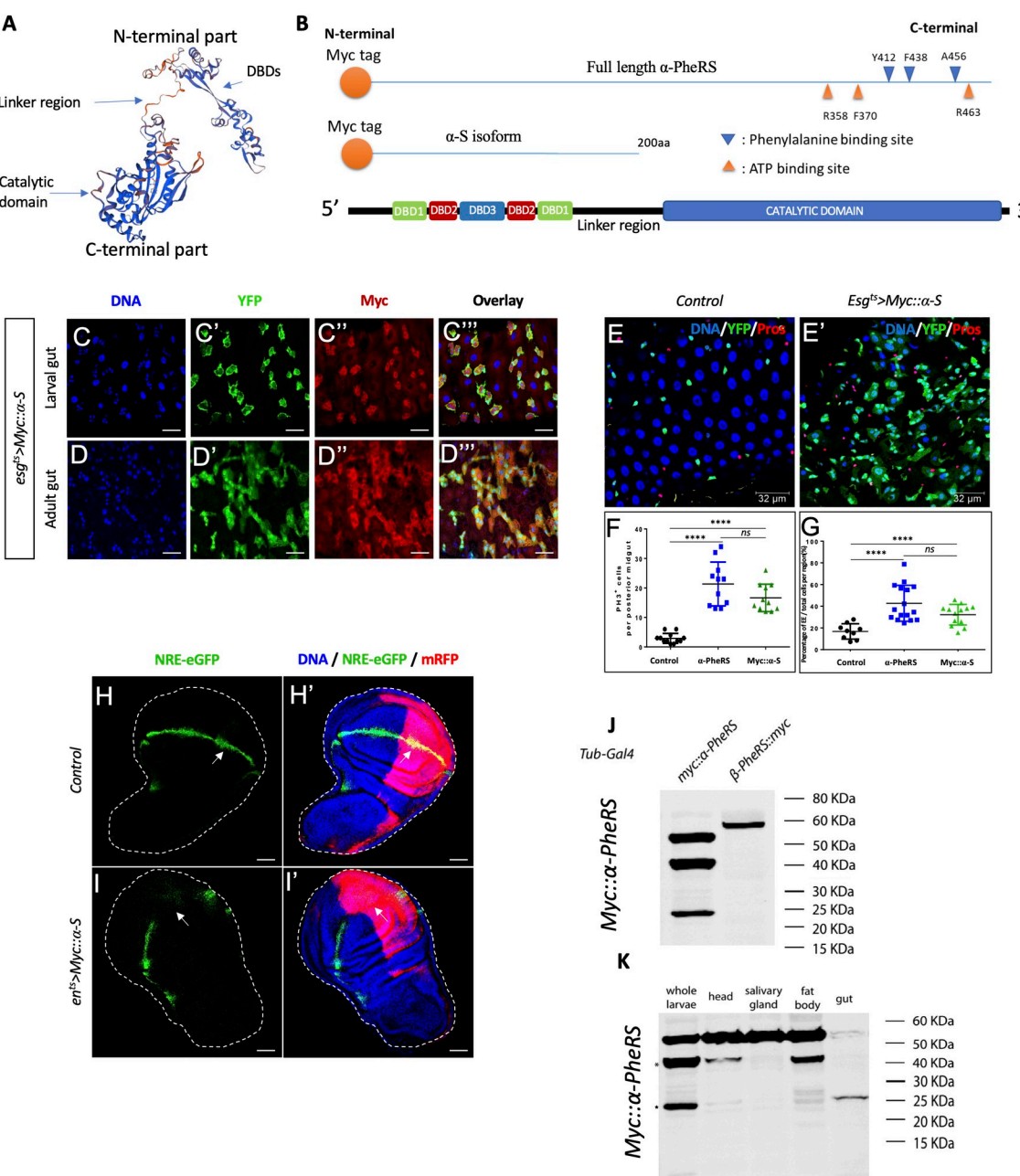

**Fig 6. The N-term of α-PheRS (α-S) is sufficient to induce the proliferative phenotype and repress Notch activity.** (A) 3D model of the α-PheRS structure (suggested by SWISS MODEL tool https://swissmodel.expasy.org/) also shows that the catalytic module and the winged DBDs are separated from each other by a linker strand. (B) Schematic illustration of the α-PheRS polypeptide with its ATP and phenylalanine binding residues. α-PheRS has the catalytic domains located in the C-terminal part and the DNA binding domains (DBD-1,2,3) located in the N-terminal region. The Myc tagged α-S construct lacks all core domains of α-PheRS but encodes the first 200 codons of the α-PheRS ORF under UAS control. (C,D) *Myc::α-S* was expressed using the esg[ts] system in adult and larval midguts and this led to similar phenotypes as elevated expression of α-PheRS[(Cys)] in the same system *(esg[ts]/UAS-Myc::α-S)*. (E-E',F,G) Quantification of PH3[+] cell numbers (F) and the proportion of EE per total cell number (G) in the intestinal region analyzed. In all experiments, EEs were visualized with the anti-Prospero antibody, YFP[-]/Hoechst[+] polyploid cells were counted as ECs. Animals were mated at 18˚C, and adults of the required genotypes were collected and shifted to 29˚C to inactivate Gal80[ts]. Adult midguts were dissected from female flies after 5 days of induction at 29˚C. At least 10 guts were analyzed for each genotype. n = 10, [**]p<0.01, [***]p<0.001, [****]p<0.0001 in ANOVA tests. (H, I) Elevated Myc::α-S leads to the loss of NRE-eGFP expression in the posterior compartments, similar to the effect of α-PheRS[(Cys)]. The same system and experimental setups were used as in Fig 5B–5E *(w, UAS-Dcr2/+;en-Gal4,UAS-mRFP,NRE-eGFP/+;tub-Gal80[ts]/UAS-Myc::α-S)*. (J, K) Besides the full-length α-PheRS isoform (55kDa), one stable Myc-tag containing isoform around 40 KDa and another around 25KDa appear when expressing transgenic Myc::α-PheRS

under the control of the *Tub-Gal4* driver *(w; +/+; Tub-Gal4/UAS-Myc::α-PheRS)*. The 40 KDa isoform was present in larval heads and fat bodies, while the 25 KDa isoform was predominantly expressed in larval guts.

Notch signaling, too, we expressed α-S with the esg$^{ts}$ driver in larval and adult guts and found that it produced an excessive number of Esg$^+$ cells (Fig 6C–6C''' and 6D–6D'''), an increase in mitotic cells (Fig 6F) and that it diverted ISC differentiation toward the EE fate (Fig 6E, 6E' and 6G). The truncated isoform α-S is capable of inducing the production of many cells with stem cell characteristics and it directs their differentiation towards an EE fate, just like α-PheRS$^{(Cys)}$ does. Similarly, expressing α-S in the posterior compartment of wing discs induced ectopic vein formation between the L4 and L5 veins in the adult wing (S4E and S4F Fig) and the same loss of the Notch reporter signal (NRE-eGFP) in the posterior compartments of wing discs as full-length *α-PheRS$^{(Cys)}$* did (Fig 6H–6I'). These observations demonstrate that elevated *α-S* levels downregulate *Notch*, mapping the inhibitory activity to the N-terminal 200 amino acids.

Expressing transgenic Myc::α-PheRS under the control of the *Tub-Gal4* driver revealed beside the full-length α-PheRS isoform (55kDa) also one stable isoform around 40 KDa and another around 25KDa (Fig 6J). To study the expression pattern of these isoforms, we dissected different larval tissues and found that the 40 KDa isoform was present in larval heads and fat bodies, while larval guts showed predominantly the 25 KDa isoform (Fig 6K). The 25 KDa isoform might either arise if Myc::α-PheRS is cleaved in the linker domain or if the pre-mRNA is processed in a different way to give rise to a shorter, but stable isoform. Because there is no evidence for alternatively processed mRNAs of α-PheRS in wild-type animals (flybase.org version FB2019_04; [45]), we tested whether PheRS might get cleaved *in vivo* to produce an isoform that resembles Myc::α-S. Myc-tagged isoforms were isolated from total larvae by immunoprecipitation and gel purification, and their polypeptides were identified by mass spectrometry (MS). The MS data shows the peptide coverage of the 25KDa isoform according to the score of Peptide Spectrum Matches (PSM; S3 Fig). The 25 KDa isoform contains the peptides of the N-terminal 28% of the full-length α-PheRS. Even though we identified two more C-terminal peptides, which could not be rejected based on the PSM interpretations, these two peptides were likely recovered due to a "memory effect" of the column. This indicates that the 25 KDa isoform is a C-terminally truncated version of α-PheRS that strongly resembles the α-S that performs the non-canonical functions.

## Mechanism of downregulating Notch signaling by α-S

Given the competitive interaction between *α-PheRS* and *Notch* and the effect of the α-S isoform on downregulating N activity, we aligned the polypeptide sequence of α-S and Notch, and this revealed similarities between α-S and the RAM domain (RBP-Jκ-associated molecule) of Notch (Fig 7A). To transmit the Notch activity, the RAM domain of the NICD binds through the tetra-peptide motif ΦWΦP to the beta-trefoil domain (BTD) of Suppressor of Hairless (Su(H)), the *Drosophila* CSL/RBPJ. This allows the complex to interact with specific DNA sequences in the promoter region of the target genes to activate transcription [46–50]. Mammals can repress Notch activity with proteins containing the tetra-peptide motif ΦWΦP. These compete with NICD to bind to CSL/RBPJ and form the Complex of Repressor (CoR) that inhibits Notch target gene activation [51–54]. Even though the tetrapeptide is only partially conserved in α-S, the similarities between α-S and the RAM domain of Notch, and the presence of the tryptophan residue (W) flanked by Φ (Fig 7A) led us to hypothesize that α-S and NICD might compete for a common target.

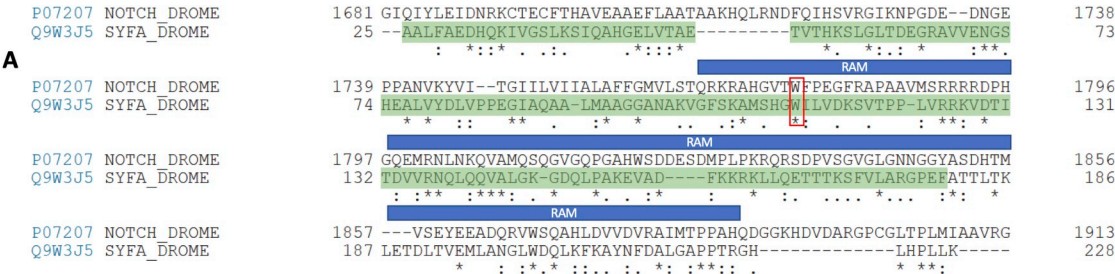

RAM domain: 1766-1896; N-terminal of α-PheRS: 1-180

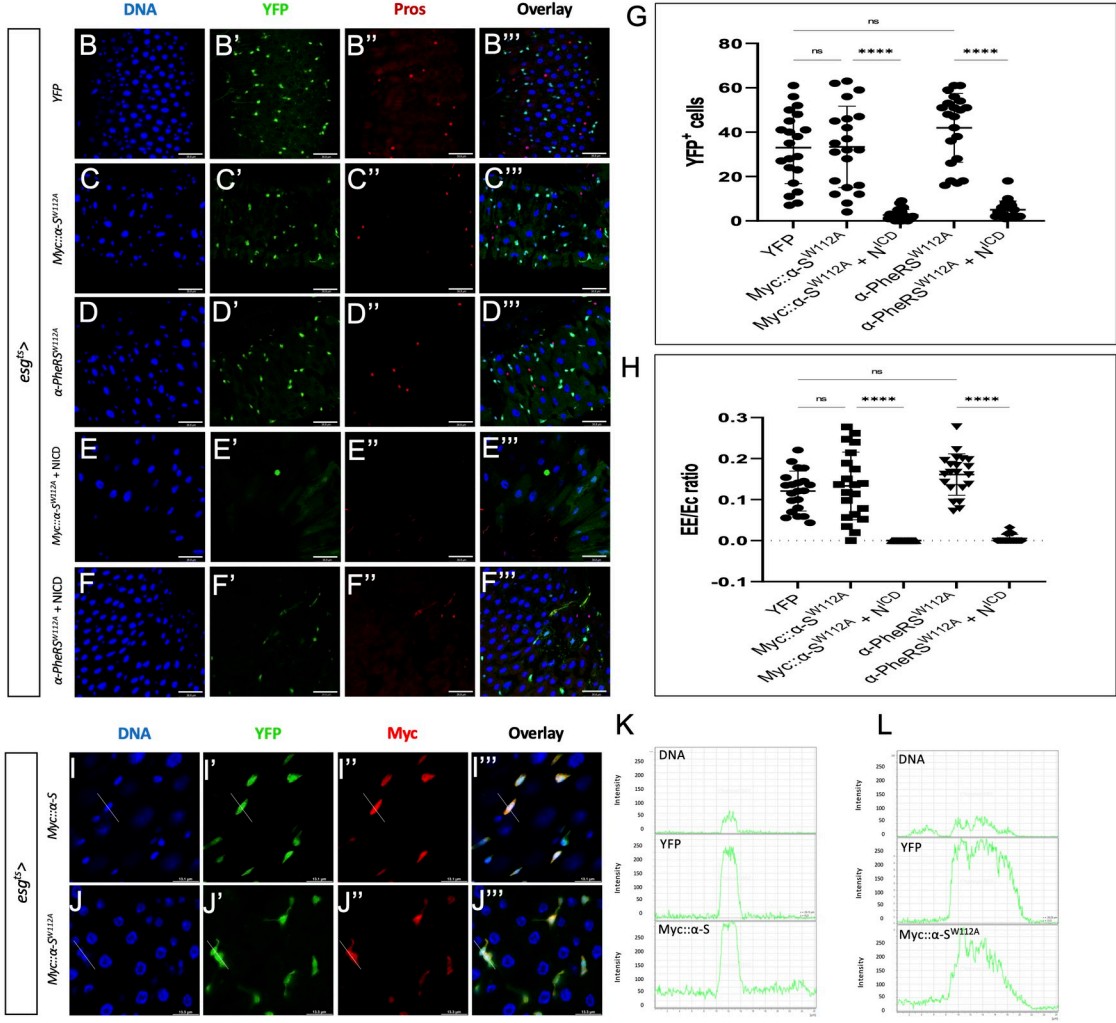

**Fig 7. Possible mechanism of the competition between N$^{ICD}$ and the α-S activity.** (A) The alignment of Notch and α-PheRS shows that moderately conserved motives are present in the RAM domain (1766–1896 aa) and the N-terminal sequence of α-PheRS (1–180 aa). The DNA binding domain (DBD) is shown shaded in green. The alignment task was conducted by the Alignment Tool of the Uniprot website (https://www.uniprot.org/) with two FASTA polypeptide sequences of *Drosophila* Notch and *Drosophila* α-PheRS. $^*$ (asterisk) indicates positions that have a single, fully conserved residue. ":" (colon) indicates conservation between groups of strongly similar properties—scoring > 0.5 in the Gonnet PAM 250 matrix. A "." (period) indicates conservation between groups of weakly similar properties—scoring = < 0.5 in the Gonnet PAM 250 matrix. (B-D''') Elevated YFP (control), Myc::α-S$^{W112A}$ or α-PheRS$^{W112A}$ levels induced by the esg$^{ts}$ system (see Fig 1E for details) did not lead to additional YFP-positive cells and also not to the hyper-accumulation of EEs. (E- F''') Neither elevated Myc::α-S$^{W112A}$ nor α-PheRS$^{W112A}$ levels rescued the depleted ISC pool caused by Notch over-activation (induced by expressing the NICD under the same control). Note that F shows only 2 Pros$^+$ cells, the weaker

signals are caused by autofluorescence from some fibers. (G) Quantification of YFP$^+$ cell numbers and (H) Proportional contribution of EEs to the total number of gut cells in the intestinal region analyzed (visualized with the 63x objective and Leica SP8). Animals were mated at 18˚C, and adults of the required genotypes were collected and shifted to 29˚C to inactivate Gal80$^{ts}$. Adult midguts were dissected from female flies after 5 days of induction at 29˚C. At least 10 guts were analyzed for each genotype. n = 10, $^{**}$p<0.01, $^{***}$p<0.001, $^{****}$p<0.0001 in ANOVA tests. (I-J''') The anti Myc staining revealed the presence of both Myc::α-S and Myc::α-S$^{W112A}$ in nuclei. Signal intensity values in the graphs (L) and (K) were measured from left to right along the lines in I''' and J''', respectively.

To test this hypothesis, we mutated the conserved "W" (tryptophane) codon in *α-S* to give rise to *Myc::α-S$^{W112A}$* and analyzed the effect of this mutation on Notch signaling in vivo. To test whether the mutation W112A does not affect the stability of the α-S polypeptide, we used the en-Gal4 system to overexpress *Myc::α-S* and *Myc::α-S$^{W112A}$* and stained larval wing discs with Myc antibodies. The mutant still showed a similar expression in the posterior compartment of the wing discs as the wild-type α-S (S2 Fig), confirming that the mutation does not affect protein stability. In contrast to what was seen upon wild-type expression of α-PheRS (Fig 2), expressing *Myc::α-S$^{W112A}$* with the esg$^{ts}$ driver in adult guts did neither induce the overproliferation phenotype (excessive numbers of Esg$^+$ cells) nor the differentiation phenotype (higher EE/EC ratio) (Fig 7B, 7C, 7G and 7H). These results could also be confirmed when a full-length version of *α-PheRS* with the mutation W112A was expressed with the same driver (Fig 7D, 7G and 7H). To further test whether α-PheRS with the W112A mutation lost its ability to interfere with Notch signaling, we simultaneously elevated levels of Myc::α-S$^{W112A}$ or α-PheRS$^{W112A}$ together with NICD levels. Indeed, the mutated Myc::α-S$^{W112A}$ or α-PheRS$^{W112A}$ did not rescue the ISC pool or the EEs, the two cell types that are depleted by NICD over-expression (Fig 7F–7F''', 7G and 7H). Furthermore, because a genomic construct expressing this mutant version of *α-PheRS$^{W112A}$ (gα-PheRS$^{W112A}$)* under its native promoter rescued the *α-PheRS* null mutant, the W112A mutation does not prevent the canonical activity of *α-PheRS* and produces sufficient protein that can perform the aminoacylation reaction. This also shows again that this mutant protein is stable.

NICD is known to enter the nucleus the perform its function in transcriptional control. To test whether α-S is also able to enter the nuclei, we use the esg$^{ts}$ system to drive Myc::α-S expression and stained guts for Myc. As shown in Fig 7I–7I''' and 7K (compare to Fig 1B and 1D), we found high levels of Myc::α-S signal in nuclei of progenitor cells (YFP$^+$ cells), revealing that α-S can translocate to nuclei. Similarly, the Myc::α-S$^{W112A}$ signal was also present in nuclei of YFP$^+$ cells (Fig 7J–7J''' and 7L), indicating that the W112A mutation neither affects stability nor translocation into nuclei. These results, therefore, show that W$^{112}$ in α-PheRS plays an important role in counteracting Notch signaling in ISCs and EBs.

## Discussion

Notch signaling regulates diverse cellular behaviors during tissue growth in a context-dependent manner. Most prominently, these are proliferation and differentiation [55,56]. In this study, we presented the effect of α-PheRS on downregulating Notch signaling activity in different tissues where Notch induces diverse outcomes. In the larval brain, Notch signaling promotes type II NB self-renewal, and ectopic expression of NICD leads to tumor formation [41–43]. Here, elevated α-PheRS showed an inhibitory effect on Notch signaling, preventing type II NB self-renewal and thereby reducing the size of the brain lobe (Fig 4). Accordingly, elevated α-PheRS in this situation also rescued the tumor phenotype caused by the expression of the activated form of Notch, NICD (Fig 4D, 4F, 4G, 4L–4N). This activity of counteracting Notch signaling maps to the N-term of α-PheRS because expressing only the N-terminal part of α-PheRS, the region that does not contain any catalytic domains and that does not interact with the β-PheRS subunit (Fig 6C, 6D, 6H, 6I; [57]) was just as effective in producing the same phenotypes as α-PheRS$^{(Cys)}$.

In the intestine, the regulation of Notch is crucial for ISC fate decisions. Any interference with this signaling pathway alters the composition of the gut cell population. We found that even modest elevation of the levels of the essential cellular household enzyme α-PheRS drastically changed the composition of the intestinal cell population. Aside from hyperaccumulation of ISCs, EE cells and "differentiating EBs" become more abundant. This phenotype has also been reported in the studies by Korzelius and colleagues [58,59]. They reported that the WT1-like transcription factor Klumpfuss (Klu) is regulated by Notch signaling and maintains the lineage commitment of enterocyte progenitors in the *Drosophila* intestine. Similar as elevated α-PheRS levels, loss of *klu* function also leads to blockage of EC differentiation, causing the "differentiating EB" phenotype and diverting EBs into EE differentiation. Because Notch signaling has been reported to regulate *klu*, it appears that α-PheRS levels also act upstream of *klu*.

Elevated levels of α-PheRS and α-S reproduced the "vein formation" phenotype in the adult wings caused by loss-of-Notch (S4 Fig) [44], and these increased levels also prevented the transcription of the Notch transcriptional reporter NRE-eGFP at the D/V boundary of larval wing discs (Figs 5B–5D, 6H, 6I). This inhibition of Notch signaling must therefore happen at or before the transcription of the reporter. Reminiscent of its function in type II neuroblasts, Notch signaling at the D/V boundary region of the wing disc promotes proliferation [60]. Upon additional expression of α-PheRS in this region, not only Notch activity (Fig 5), but also PH3+ cells became absent from this region [14]. Interestingly, however, in regions of the posterior wing disc compartment where there is no Notch activity, elevated α-PheRS caused the appearance of additional pH3+ cells [14]. Because knockdown of *N* causes the notched wing phenotype but increased α-PheRS levels do not (S4 Fig), it would be interesting to find out whether the proliferation of these additional pH3+ cells that reside outside of the region with *Notch* activity can compensate for the cells along the D/V boundary that depend on Notch signaling to be mitotically active. We do not know yet how elevated α-PheRS levels induce this cell proliferation, but an effect on the organ size control pathway Hippo signaling might be a candidate. On the other hand, the complexity of the Notch signaling at the D/V boundary, where downregulation or overactivation of Su(H) can cause indistinguishable phenotypes [20] makes conclusions difficult. Furthermore, cell proliferation in the distal wing margin is regulated also by epidermal growth factor (EGF) or decapentaplegic (Dpp)/bone morphogenetic protein (BMP) signaling [61–63], suggesting that interactions with other signaling mechanisms might be involved, too. In any case, we conclude that even in the wing, α-PheRS represses Notch signaling, but different regions of the wing disc respond differently to α-PheRS levels, making the wing disc a more complex system to study the responses to elevated α-PheRS levels.

In ovarian follicle cells, the elevated levels of α-PheRS led to an increase in clone size with more cells per clone [14]. This could possibly also be caused by an inhibitory effect of α-PheRS on Notch signaling because Notch is required for the follicle cells to exit the mitotic cycle and switch to the endocycle [64] and loss of Notch signaling prevents the epithelial cells from switching from the mitotic cell cycle to the endocycle, leading to over-proliferation of follicle cells [65,66]. Similarly, in adult midguts, elevated α-PheRS interfered with normal homeostasis of ISCs by inducing a typical low Notch signaling phenotype with many more ISCs, EBs, and EEs [22,23,67]. Because α-PheRS levels attenuate Notch signaling in different situations where Notch causes diverse effects, α-PheRS (or a part of it) seems to be a general component of the *Drosophila* Notch signaling pathway.

Repression of Notch signaling through competition with the NICD has been described in different systems and has recently been reviewed [68,69]. Best studied is probably the competition involving the factors binding to the C-terminal domain (CTD) of Su(H) (CSL in vertebrates and RBPJ in mammals) that lead to the formation of the Complex of Repressor (CoR)

that inhibits Notch target gene activation. A less well-studied group of competitors compete directly with the binding of the RAM domain of the NICD to the BTD of the Su(H) ortholog CSL/RBPJ [68,69]. This competition involves the 20 N-terminal amino acids of the RAM domain, which contain the hydrophobic tetrapeptide with the key tryptophane residue (ΦWΦP). So far, this competition was only seen in mammals and, therefore, considered to be specific for vertebrate Notch signaling. Known competitors are Epstein-Barr nuclear antigen 2 (EBNA2), mouse KyoT2/FHL1, and human RITA (RBPJ interacting and tubulin associated) [70–75]. Based on our results, a naturally occurring fragment of the α-PheRS household protein might function through this mechanism to modulate or attenuate Notch signaling if this invariant W is present. The immediately adjacent hydrophobic residues are also present in the α-S tetrapeptide sequence. However, the residue in position four, a proline, is substituted by another hydrophobic residue, leucine. Interestingly, even in mammals, this residue is less conserved than the W in position two [69]. Two additional elements have been discussed to affect the affinity of the interaction between the RAM domain and the transcription factor (CSL, RBPJ, or Su(H)) [68]. The dipeptides -HG–and–GF–. Interestingly, both dipeptides are also present in the N-terminal 20 amino acids of the RAM domain of *Drosophila* Notch, α-PheRS, and α-S even though they are found in different positions (Fig 7A). Their contribution to the binding affinity would therefore have to be further evaluated. Additionally, the α-S polypeptide also contains a KKRK sequence element that might function as a nuclear localization sequence (NLS). It could therefore allow α-S to actively enter the nucleus to bind to a partner (Fig 7I– 7I'''). Notably, α-PheRS(Cys) is a cytoplasmic protein for which we did not observe clear nuclear localization (Fig 1B and 1D). This might suggest that processing or truncation of α-PheRS(Cys) facilities the translocation of its signaling domain α-S into nuclei, a process that might even play an important role in fine-tuning this novel α-PheRS activity. This model is also consistent with the result that α-PheRS and α-S act on Notch signaling at or before the transcription from the Nre element of the Nre-eGFP reporter and, finally, an α-S activity in transcriptional control could also shed light on the puzzling question of why a cytoplasmic aaRS (i.e. α-PheRS) contains a DNA binding domain. However, at least until *in vivo* binding of α-S to Su (H) is demonstrated, other competitions with Notch still need to be considered. Furthermore, because α-S also displays proliferative activities that seem not to be mediated by its effect on Notch signaling, we expect that α-S or its downstream targets act also on components of other signaling pathways, like for instance the organ size control pathway.

A correlation between elevated levels of α-PheRS/FARSA and tumor formation has been noted some time ago [12] and is also suggested by the data published in the GENT2 database [11]. Not the least because colon cancers are among the ones that express higher levels of α-PheRS in the tumor tissue than in the healthy counterpart, modeling the effect of elevated α-PheRS levels in *Drosophila* ISCs and EBs might reveal mechanisms contributing to tumor development. In the gut tissue, elevated α-PheRS and α-PheRS(Cys) levels produced 12–14 times as many tumors and these showed mostly a more severe phenotype (Fig 3A–3F). Such tumors were composed of ISCs and differentiating EBs. Similarly, guts also contained 5–8 times as many mitotic cells when α-PheRS(Cys) levels were elevated (Fig 3L). Together, these results strongly suggest that higher levels of α-PheRS can induce strong cell proliferation. Additionally, because these over-proliferation cells display stem cell characteristics and changes in cell fate, elevated α-PheRS levels might be a risk factor for tumor formation.

The hyperaccumulation of cells with stem cell characteristics is mediated by high α-PheRS(Cys) or α-S repressing Notch signaling and possibly affecting also another signaling pathway. In mammals, Notch signaling is essential for maintaining the homeostasis of cell proliferation and differentiation [76] similar to the function of Notch signaling in the *Drosophila* gut that is needed to prevent the induction of tumors in the adult midgut [22,23]. In humans,

misregulation of Notch signaling in these processes has been suggested to trigger the development of colon cancer, and *Notch* has been proposed as a molecular target for cancer therapy [77]. Furthermore, several other tumors contain low Notch levels and elevated α-PheRS/ FARSA levels [11]. Amongst them are for instance skin, head and neck, soft tissue, muscle, and tongue tumors. In these cases, it would be interesting to find out whether elevated FARSA levels cause the downregulation of Notch. Notch signaling has been found inactivated in different squamous cell carcinoma, including cutaneous, head and neck, and esophageal squamous cell carcinoma, and also in small-cell lung cancers (ScLc) [78]. However, Notch signaling in tumors is more complex because Notch acts intrinsically both tumor suppressive and oncogenic. The latter has been observed in some subtypes of gastric and esophageal cancers, colorectal cancer, uterine corpus endometrial cancer, breast cancer, and non-small-cell-lung cancer [78]. A more in-depth analysis of the relationship between FARSA levels and Notch activity should therefore also consider that elevated levels of FARSA might reflect a reaction of the cell to counteract excessive Notch activity. The results presented here provide new and unexpected insights into the regulation of Notch signaling in the context of gut tumorigenesis and they suggest new opportunities to target these mechanisms.

## Materials and methods

### Fly genetics and husbandry

All *Drosophila melanogaster* fly stocks were kept for long-term storage at 18˚C in glass or plastic vials on standard food with day/night (12h/12h) light cycles. All experiments were performed at 25˚C unless specifically mentioned. A UAS-GFP element was added in experiments that tested for rescue and involved Gal4-mediated expression of the rescue gene. This construct served to even out the number of UAS sites in each Gal4 expressing cell. Origins of all stocks are noted in the Table 1 (*Key Resource Table)*.

### DNA cloning and generation of transgenic flies

Sequence information was obtained from Flybase. All mutations and the addition of the Myc-tag to the N-terminus of *α-PheRS* were made by following the procedure of the QuickChange Site-Directed Mutagenesis Kit (Stratagene). The genomic *α-PheRS* rescue construct (*Myc*::*α-PheRS*) codes for the entire coding region and an additional Myc tag at the N-terminal end. In addition, it contains ~1kb of up-stream and down-stream sequences and it was cloned into the $pw^+SNattB$ transformation vector [16,79]. The *α-PheRS* and *β-PheRS* cDNAs were obtained by RT-PCR from mRNA isolated from 4–8 days old *OreR* flies [16]. Transgenic flies were generated by applying the $\phi$ C31-based integration system with the stock (*y w att2A[vas-ϕ]; +; attP-86F*) [80]. Kits, vectors, bacterial strains, buffers and primers are described in Tables 1–3.

### Western blotting

Protein was extracted from tissues, whole larvae, or flies using lysis buffer. 25 guts were analyzed for each genotype. Protein lysates were separated by SDS-PAGE and transferred onto PVDF membranes (Milipore, US). The blocking was performed for 1h at room temperature (RT) with non-fat dry milk (5%) in TBST solution. Blots were probed first with primary antibodies (diluted in blocking buffer) overnight at 4˚C and then with secondary antibodies (diluted in TBST) 1h at RT. The signal of the secondary antibody was detected by using the detect solution mixture (1:1) (ECL Prime Western Blotting System, GE Healthcare Life Science) and a luminescent detector (Amersham Imager 600, GE Healthcare Life Science). Origins and recipes of all buffers and reagents are noted in Tables 1 and 2.

**Table 1. Key Resources Table.**

| Reagent or Resource | Sources | Identifier | Additional information |
|---|---|---|---|
| **Antibodies** | | | |
| Anti phospho-Histone H3-rabbit | Cell signaling | 9701S | 1:200 v/v |
| Anti phospho-Histone H3-mouse | Cell signaling | 9706S | 1:200 v/v |
| Anti α-PheRS | Genescript | 4668 | Customized product (1:200 v/v) |
| Anti α-PheRS | Genescript | 4669 | Customized product (1:200 v/v) |
| Anti Myc-mouse | Developmental Studies Hybridoma Bank (DSHB) | 9E10 | Supernatant (1:3 v/v) |
| Anti Puromycin | DSHB | PMY-2A4 | 1:100 v/v |
| Anti Prospero | DSHB | MR1A | 1:200 v/v |
| Anti Delta | DSHB | C594.9B | 1:10 v/v |
| Anti V5 tag-rabbit | Cell signaling | 13202 | 1:200 v/v |
| Anti Cy3 rabbit | Jackson Immuno Research | 115-165-146 | 1:200 v/v |
| Anti-rabbit Alexa Flour 488 | Molecular Probes | A-11008 | 1:200 v/v |
| Anti-rabbit Alexa Flour 488 | Molecular Probes | A-11034 | 1:200 v/v |
| Anti-mouse Alexa Flour 488 | Molecular Probes | A-11029 | 1:200 v/v |
| Anti-rabbit Alexa Flour 488 | Life technology | A-21206 | 1:200 v/v |
| Anti-rabbit Alexa Flour 594 | Invitrogen | A-11037 | 1:200 v/v |
| Anti-mouse Alexa Flour 594 | Molecular Probes | A-11032 | 1:200 v/v |
| Anti-mouse Alexa Flour 568 | Life technology | A-10037 | 1:200 v/v |
| Anti α-tubulin | Abcam | Ab18251 | 1:1,000 v/v |
| Anti GFP | ImmunoKontact | 042704 | 1:1,000 v/v |
| Anti Myc-rabbit | Santa Cruz | Sc-789 | A-12 (1:1,000 v/v) |
| HRP anti-rabbit IgG antibody (Peroxidase) | Vector | PI-1000 | 1:10,000 v/v |
| HRP anti-rabbit IgG antibody (Peroxidase) | Vector | PI-2000 | 1:10,000 v/v |
| **Fly stocks and genetics** | | | |
| α-PheRS$^{G2060}$/FM6 | Bloomington *Drosophila* Stock Center (BDSC) | 26625 | |
| gα-PheRS$^{Cys}$ <br> UAS-α-PheRS$^{Cys}$ | | | Transgenic construct <br> Transgenic construct |
| hspFLP; Act-Gal4/CyO; neoFRT82B, tub-Gal80/TM3, Sb | | | |
| w; If/CyO; neoFRT82B, UAS-α-PheRS$^{(Cys)}$ | | | |
| engrailed-Gal4 | BDSC | 30564 | |
| engrailed-Gal4, NRE-EGFP, UAS-myrRFP | BDSC | 30730 | |
| UAS-GFP | BDSC | 6658 | |
| w; UAS-Myc::MYC | BDSC | 9674 | |
| hspFLP/y; +; UAS-Myc::MYC | BDSC | 9675 | |
| hspFLP/y; UAS-NICD /CyO; MKRS/TM2 | BDSC | 52008 | |
| NRE-EGFP | BDSC | 30727 | |
| UAS-N RNAi | BDSC | 7078 | |
| neoFRT82B Sb1/TM6 | BDSC | 2051 | |
| NRE-eGFP | BDSC | 30728 | |
| UAS-Dl | BDSC | 5614 | |
| tub-Gal4/TM3, Sb | BDSC | 5138 | |
| w$^*$, Insc-Gal4 | BDSC | 8751 | |
| w$^*$, UAS-Dicer2; wor-Gal4, ase-Gal80; UAS-mCD8::GFP | IMBA | | A gift from Juergen A. Knoblich, IMBA |
| y w att2A[vas-φ]; +; attP-86F | ETH Zurich | | A gift from Hugo Stocker, ETH |

*(Continued)*

**Table 1.** (Continued)

| Reagent or Resource | Sources | Identifier | Additional information |
|---|---|---|---|
| *esg-Gal4, UAS-2XEYFP; MKRS/TM6B, Tb* | ETH Zurich | | A gift from Hugo Stocker, ETH |
| *esg-Gal4, UAS-2XEYFP; tub-Gal80ᵗˢ/TM3, Sb* | ETH Zurich | | A gift from Hugo Stocker, ETH |
| *yw; esg-Gal4, UAS-GFP/TM6B, Tb, Hu* | ETH Zurich | 2400 | A gift from Hugo Stocker, ETH |
| *NP1-Gal4 (Myo31DF or Myo1A-Gal4)/CyO, y+* | ETH Zurich | 2398 | A gift from Hugo Stocker, ETH |
| *esg-Gal4, UAS-mCherry-CD8, tub-gal80ᵗˢ/ CyO* | | | A gift from Péter Nagy, Cornell University |
| *yw;UAS-cyto-gars-myc/CyO* | | | A gift from Albena Jordanova, VIB-U Antwerp Center for Molecular Neurology |
| **Bacteria strains and vectors** | | | |
| XL1 blue | Aligent | 200249 | |
| Rosetta–Novagen | Merck milipore | 70954 | |
| pET-28a –Novagen | Merck milipore | 69864 | |
| pET LIC (2A-T) | Addgene | 29665 | |
| pUASattB | *Drosophila* Genomics Resource Center | 1419 | |
| pw+SNattB | [79] | | |
| **Commercial assay or kit** | | | |
| Pierce Silver Stain kit | Thermo Scientific | 24612 | |
| Pierce BCA Protein Assay kit | Thermo Scientific | 23227 | |
| ReliaPrep DNA CleanUp and Concentration System | Promega | A2893 | |
| GeneElute HP Plasmid miniprep kit | Sigma | NA0160 | |
| Qiagen Plasmid Plus Midi kit | Qiagen | 12943 | |
| Ni-NTA affinity resin | Qiagen | 30210 | |
| ECL Prime Western Blotting System | GE Healthcare | RPN2232 | |
| RNAMaxx High Yield Transcription Kit | Agilent | 200339 | |
| **Software, algorithm** | | | |
| Leica Application Suite X (LAS X) | Leica | | https://www.leica-microsystems.com/products/microscope-software/p/leica-las-x-ls/ |
| FIJI | ImageJ | | https://fiji.sc/ |
| GraphPad Prism | GraphPad | | https://www.graphpad.com/scientific-software/prism/ |
| FlowJo | BD Biosciences | | https://www.flowjo.com/ |
| Microsoft Excel | Microsoft | | https://products.office.com/en-us/excel |

## Immunofluorescent staining and confocal microscopy

Adult midguts were dissected from each female fly after 5 days at 29˚C (unless stated otherwise) or at different time points after 2, 3, 4 days of heat treatment for the time-course experiment. A total of 10 guts were analyzed for each genotype. In the larval experiments, the animals were mated at 25˚C on fresh, rich diet food and set up for egg-laying during a 2-hour window. 120 hours after egg-laying, larval midguts were dissected from wandering third instar larvae and a total of 10 guts were analyzed for each genotype. Dissections were performed in 1X PBS on ice and tissues were collected within a maximum of one hour. Fixation with 4% PFA in PBS-T 0.2% at RT was done for different durations depending on the different tissues: two hours (guts), 40 minutes (brains), 30 minutes (wing discs, ovaries). Then the samples were blocked overnight with a blocking buffer at 4˚C. Primary antibodies (diluted in blocking buffer) were incubated with the samples for 8h at RT. The samples were rinsed 3 times and washed 3 times (20 minutes/wash) with PBST. Secondary antibodies (diluted in PBST) were

**Table 2. Buffers.**

| Lysis buffer for *Drosophila* tissue | Lysis buffer for bacteria |
|---|---|
| 20 mM Tris HCl pH7.4<br>150uM NaCl<br>2 mM EDTA<br>50 mM NaF<br>10% Glycerol<br>1% Triton X100<br>1 Protease inhibitor cocktail tablet<br>(Roche-4693159001)<br>1 mM phenylmethylsulphonyl fluoride | 20 mM Tris HCl pH7.4<br>150uM NaCl<br>2 mM EDTA<br>50 mM NaF<br>10% Glycerol<br>1% Triton X100<br>4mM Imidazole 1M<br>0.6% Lysozyme<br>1 Protease inhibitor cocktail tablet<br>1 mM phenylmethylsulphonyl fluoride |
| **4% PFA** | **1X PBST** |
| 1X PBST<br>4% (w/v) Paraformaldehyde | 0.2% (v/v) Tween 20<br>1X PBS |
| **Blocking buffer** | **Fly food recipe** |
| 5% (w/v) non-fat dry milk<br>0.1% (v/v) Triton X100 | 20.4 l $H_2O$<br>1,680 g Maize flour<br>720 g Yeast<br>1,800 g Syrup<br>192 g Potassium sodium tartrate tetrahydrate<br>36 g Nipagin<br>120 ml Propionic acid |
| **10X PBS pH 7.4** | |
| 10.6 mM $KH_2PO_4$<br>1.5 M NaCl<br>30 mM $Na_2PO_4.7H_2O$ | |
| **10X SDS running buffer** | **10X Transfer buffer** |
| 30 g Tris base<br>144 g Glycine<br>10 g SDS<br>$dH_2O$ to 1 L | 30 g Tris base<br>144 g Glycine<br>$dH_2O$ to 1 L |
| **10X TBS pH to 7.6** | **1X TBST** |
| 24 g of Tris Base<br>88 g of NaCl<br>$dH_2O$ to 1 L | 100 mL 10X TBS<br>900 mL $dH_2O$<br>0.1% (v/v) Tween 20 |

incubated overnight at 4˚C. The samples were then rinsed 3 times and washed 2 times (20 minutes/wash) with PBST. Hoechst 33258 (2.5 µg/ml) was added in PBST before the third and last washing step and the samples were mounted with Aqua/Poly Mount solution (Polysciences Inc., US). For the anti-Delta labeling, the samples were blocked for 3h at RT with a

**Table 3. Primers.**

| Name | Sequence (5' to 3') | Application |
|---|---|---|
| rc2263f | CGCGGATCCATCCGGCGAGAGAGTGTCTTTG | Genomic genomic construct of *α-PheRS* |
| rc2263r | CGGGGTACCTATGCCTGGCGATAATCGTG | |
| Tyr412Cys & Phe438Cys-F | TCAAGCCGGCGTACAATCCGTGTACCGAGCCCAG | Construct of *α-PheRS*$^{Cys}$ mutation |
| Tyr412Cys & Phe438Cys-R | CTCCGGCCGACAGACGCCCGAGTTGCCC | |
| seq r6 | GCTCCCATTCATCAGTTCC | Sequencing |
| seqA r1 | CATTTCCACCGTGAGATCCGTC | Sequencing |
| seqA r2 | AACTCTTGTGGGTGACCGTTTC | Sequencing |
| seqA f1 | GTTCTCGAAGTGAATGTTCTGG | Sequencing |
| seqA f2 | TTTAGCCACCGTCGTCGTTTC | Sequencing |
| seqA r3 | TCCAGCGACGATGACGAATTTG | Sequencing |
| seqA f3 | CAAATGGATTGTGGGACCAGC | Sequencing |
| seqA r4 | GCCCTCCTCCACCATCTTTAG | Sequencing |

blocking buffer. The primary anti-Delta antibody (1:10 v/v) was incubated with the samples overnight at 4°C and then the secondary antibody was incubated overnight at 4°C. Origins and diluted concentrations of all buffers and antibodies are noted in the Tables 1 and 2.

## Image acquisition and processing

Imaging was carried out with a Leica SP8 confocal laser scanning microscope equipped with a 405 nm diode laser, a 458, 476, 488, 496, and 514 nm Argon laser, a 561 nm diode-pumped solid-state laser, and a 633 nm HeNe laser. Images were obtained with 20x dry and 63x oil-immersion objectives and 1024x1024 pixel format. Images were acquired using LAS X software. The images of the entire gut were obtained by imaging at the standard size and then merging maximal projections of Z-stacks with the Tiles Scan tool. Fluorescent intensity was determined from FIJI software.

## Quantification of cell numbers per posterior midgut

Z stack images through the width of the posterior midgut were acquired along the length of the posterior midgut from the R4a compartment to the midgut-hindgut junction. Maximum projections of each Z stack were obtained, and the total numbers of each cell type were counted manually and exported to Microsoft Excel and GraphPad Prism for further statistical analysis.

## Quantification and statistical analysis

For quantifications of all experiments, *n* represents the number of independent biological samples analyzed (the number of guts, the number of wing discs, the number of twin spots), error bars represent standard deviation (SD). Statistical significance was determined using the t-test or ANOVA as noted in the figure legends. They were expressed as P values. (*) denotes $p < 0.05$, (**) denotes $p < 0.01$, (***) denotes $p < 0.001$, (****) denotes $p < 0.0001$. (*ns*) denotes values whose difference was not significant.

## Supporting information

**S1 Fig. α-PheRS or α-PheRSCys were overexpressed using the esg-Gal4,UAS-2XEYFP;tub-Gal80ts (= esgts) system that allowed us to control the expression time to study the kinetics of the appearance of the phenotypes.** The kinetics of the *Notch* knockdown resembled partially the one of *PheRS(Cys)* expression (esg^ts / UAS-α-PheRS^(Cys)). Animals were mated at 18°C, and adults of the required genotypes were collected and shifted to 29°C to inactivate Gal80^ts. Adult midguts were dissected from female flies after the indicated induction times.
(PDF)

**S2 Fig. Staining wing discs for Myc::α-S showed the stable expression of both truncated alleles Myc::α-S and Myc::α-SW112A.** The *en-Gal4,UAS-EGFP;tub-Gal80^ts* (en^ts) system was used to drive transgene expression in the posterior compartment of developing wing discs (en^ts/*UAS-Myc*::α-S or *Myc*::α-S^W112A) and Hoechst staining to label nuclei. Animals were initially kept at 18°C for 3 days and then shifted to 29°C to inactivate Gal80^ts until adult flies hatched, enabling expression of Myc::α-S or Myc::α-S^W112A.
(PDF)

**S3 Fig. The Myc-tagged isoforms from whole larvae were purified by immunoprecipitation and gel purification.** The tryptic peptide fragments of the 25kDa band were subsequently analyzed by mass spectrometry (MS). The MS data analysis revealed the peptide coverage of the 25KDa isoform according to the score of Peptide Spectrum Matches (PSM). The 25 KDa

isoform contains the peptides of the N-terminal 28% of the full-length α-PheRS.
(PDF)

**S4 Fig. The ectopic wing venation phenotype resulting from elevated levels of α-PheRS (Cys) and α-S is similar to the phenotype of NRNAi treatment.** An ectopic vein branches from the connecting vein between the L4 and L5 vein (arrowhead in B-F). D) Knockdown of *Notch* shows this phenotype and the classical notched phenotype in the distal region of L3, L4, and near the L5 vein. The *en-Gal4,tub-Gal80^ts* (en^ts) system was used to drive transgene expression in the posterior compartment of developing wing discs *(en^ts/UAS-α-PheRS^(Cys))*. Animals were initially kept at 18˚C for 3 days and then shifted to 29˚C to inactivate Gal80^ts until adult flies hatched, enabling expression of α-PheRS, α-PheRS^Cys, α-S, or *N*^RNAi.
(PDF)

## Acknowledgments

We thank Peter Nagy, Hugo Stocker, Albena Jordanova, Erik Storkebaum, and the Bloomington Stock Center for fly stocks. We are also grateful to Rohan Chippalkatti for sharing his knowledge about the larval brain.

## Author Contributions

**Conceptualization:** Manh Tin Ho, Jiongming Lu.

**Data curation:** Manh Tin Ho, Jiongming Lu, Beat Suter.

**Formal analysis:** Manh Tin Ho, Jiongming Lu, Beat Suter.

**Funding acquisition:** Beat Suter.

**Investigation:** Manh Tin Ho, Jiongming Lu, Paula Vazquez-Pianzola.

**Methodology:** Manh Tin Ho, Jiongming Lu, Paula Vazquez-Pianzola.

**Project administration:** Manh Tin Ho, Beat Suter.

**Resources:** Manh Tin Ho, Beat Suter.

**Supervision:** Paula Vazquez-Pianzola, Beat Suter.

**Validation:** Manh Tin Ho.

**Visualization:** Manh Tin Ho, Jiongming Lu.

**Writing – original draft:** Beat Suter.

**Writing – review & editing:** Manh Tin Ho, Beat Suter.

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
